# A single-molecule method for measuring fluorophore labeling yields for the study of membrane protein oligomerization in membranes

**Melanie Ernst[1], Tugba N. Ozturk[1,2], Janice L. Robertson[1]** *

**1** Department of Biochemistry and Molecular Biophysics, Washington University School of Medicine, St. Louis, Missouri, United States of America, **2** Theoretical Molecular Biophysics Laboratory, National Heart, Lung and Blood Institute, National Institutes of Health, Bethesda, Maryland, United States of America

* janice.robertson@wustl.edu

## Abstract

Membrane proteins are often observed as higher-order oligomers, and in some cases in multiple stoichiometric forms, raising the question of whether dynamic oligomerization can be linked to modulation of function. To better understand this potential regulatory mechanism, there is an ongoing effort to quantify equilibrium reactions of membrane protein oligomerization directly in membranes. Single-molecule photobleaching analysis is particularly useful for this as it provides a binary readout of fluorophores attached to protein subunits at dilute conditions. However, any quantification of stoichiometry also critically requires knowing the probability that a subunit is fluorescently labeled. Since labeling uncertainty is often unavoidable, we developed an approach to estimate labeling yields using the photobleaching probability distribution of an intrinsic dimeric control. By iterative fitting of an experimental dimeric photobleaching probability distribution to an expected dimer model, we estimate the fluorophore labeling yields and find agreement with direct measurements of labeling of the purified protein by UV-VIS absorbance before reconstitution. Using this labeling prediction, similar estimation methods are applied to determine the dissociation constant of reactive CLC-ec1 dimerization constructs without prior knowledge of the fluorophore labeling yield. Finally, we estimate the operational range of subunit labeling yields that allows for discrimination of monomer and dimer populations across the reactive range of mole fraction densities. Thus, our study maps out a practical method for quantifying fluorophore labeling directly from single-molecule photobleaching data, improving the ability to quantify reactive membrane protein stoichiometry in membranes.

## Introduction

Ion channels, transporters and membrane protein receptors are often found in higher-order oligomeric assemblies [1]. In some cases, these assemblies are essential for function like for

**Data Availability Statement:** All relevant data are within the paper and its Supporting Information files.

**Funding:** ME, TNO and JLR were supported with a grant (R01GM120260) from the National Institutes of General Medical Sciences, National Institutes of Health. The funders had no role in the study design, data collection and analysis, decision to publish, or preparation of the manuscript.

**Competing interests:** The authors have declared that no competing interests exist.

potassium channels in which the tetrameric structure confers the ion conduction pathway along the central oligomeric axis [2]. In other cases, oligomerization is observed but not strictly required. This is the case for proteins that exhibit a parallel pathways architecture where the transport pathway is contained within each subunit, like the homodimeric CLC-ec1 chloride/proton antiporter [3]. Many of the proteins that adopt parallel pathway assemblies form as dimers, trimers, tetramers and beyond, raising the question as to why such complexes are evolutionarily favored. Indeed, for some proteins, oligomerization has been linked to regulatory impact on protein function [4], and introduces the idea of membrane proteins participating in dynamic association reactions that may be tuned by physiological factors [5, 6].

Thus, the study of dynamic protein oligomerization in membranes is an area of growing importance within the field of membrane physiology. Previous studies used methods such as electron paramagnetic resonance [7], fluorescence correlation spectroscopy [8] or Förster resonance energy transfer [9] to report on changes in oligomer populations for weak affinity complexes. On the other hand, the method of single-molecule photobleaching analysis has been particularly useful for the study of stronger membrane protein complexes. In this approach, each protein subunit is labelled with a fluorophore, either by attachment of an organic dye or fusion with a fluorescent protein, e.g. green fluorescent protein (GFP) [10]. The membrane samples are imaged on a microscope capable of detecting single-molecules, such as a total internal reflection fluorescence (TIRF) microscope, and after sufficient excitation, the fluorescent probe irreversibly bleaches. By analyzing the fluorescence intensity over time, an observer can count the step-down bleaching events as a binary readout of the existence of a fluorophore, providing a rigorous output of the number of observable fluorophores in the area of membrane examined. Because of this direct readout, single-molecule photobleaching analysis has become a popular tool to examine membrane protein oligomerization in examples such as the CLC-ec1 chloride/proton antiporter [11], mechanosensitive channels [12], calcium release-activated calcium channels [13] and receptors [14] as well as larger protein assemblies such as receptor ion complexes [15] and channel auxiliary subunit complexes [16].

While it is true that photobleaching provides a direct readout of fluorophores in a particular area, this does not provide a direct readout of protein stoichiometry. In reality, for single-molecule photobleaching to quantitively report on the oligomer populations, several additional considerations must be taken. Our studies involve imaging organic Cyanine-5 (Cy5) fluorophores on a TIRF microscope allowing us to resolve single-molecules within $\approx$ 200 nm due to the diffraction limit [17]. Yet, dilute membrane conditions, e.g. 1 subunit per 100,000 lipids, correspond to greater than 1 subunit per 200 nm diameter circle, and so there is sufficient probability that two molecules will be within the diffraction limit and appear as a single spot. In this case, dissociated monomers may yield more than one step in the photobleaching trace even though the proteins are not assembled as dimers. Free diffusion of molecules in the membrane complicates the analysis further, as the protein may leave the field before step counting is completed. To address these issues, we developed the method of single-molecule subunit capture, where protein species are rapidly and irreversibly captured into liposomes following a Poisson process, and the liposomes can be diluted and spatially separated onto the microscope slide [11, 18]. While multiple protein species can be randomly captured into the same liposome, the probability of this is predictable, depending on the density of dispersed particles defined by the monomer-oligomer equilibrium and heterogeneous Poisson distribution that can be calculated using the size distribution of the liposomes, which can be obtained by cryo-electron microscopy [19, 20]. This approach allows for the quantification of protein stoichiometry distributions across a wide range of densities in the membrane.

Yet, even at dilute conditions where single-molecules prevail, there is another factor that confounds the interpretation of photobleaching data. This is whether a fluorophore reliably

reports on a single protein subunit, which depends on the fluorophore/subunit labeling yield, the specificity of the labeling reaction and the fluorophore activity. Realistically, the subunit labeling probability ($P_{fluor}$) will be less than 100%, unless there exists significant background labeling. Therefore, for a dimeric protein at dilute conditions, the binomial probability distribution predicts that a significant fraction of dimers will possess a single fluorophore making it appear as a single step. This has been observed in studies using GFP, as maturation of the fluorophore includes folding, cyclisation, oxidation and dehydration, and may be incomplete at the time of measurement or simply unproductive [21]. This can also occur when using organic fluorophores, where the conjugation reaction may be incomplete, or cysteines remain unreactive. Alternatively, it is possible that a monomer will exhibit some two-step photobleaching traces due to a small but significant amount of non-specific background labeling, $P_{bg}$, allowing subunits to have more than one fluorophore conjugated to it. For purified protein samples, the fluorophore labeling yield can be measured directly from the absorbance of the fluorophore and protein by UV-VIS spectroscopy. However, this quantification may not be accurate as correction factors may depend on environmental conditions, e.g. purified detergent micelle vs. lipid bilayer. Also, different sample conditions may enable the fluorophore to photochemically transition to non-fluorescent states during the experiment. For example, Cy5 exhibits photo-switching behavior and may enter a dark state depending on environmental or excitation conditions [22, 23]. Furthermore, the quantification of fluorescent labeling yields is even more challenging when the protein is expressed *in vivo*. The probability that a fluorescent fusion protein is mature and fluorescent must be inferred from other studies and could change depending on expression conditions in the cell [24]. Thus, there is a need for a direct quantification of fluorophore labeling yields at the time of the experiments in order to obtain accurate information about membrane protein oligomerization.

To address this, we investigated whether our single-molecule photobleaching analysis and subunit capture approach could be applied in reverse to determine the fluorophore labeling yield of a protein. Knowing the liposome size distribution and the photobleaching distribution for a defined dimer, can we estimate the subunit and background labeling yields from the experimental photobleaching data? To test this, we examined our previous data from a cysteine cross-linked dimer form of the CLC-ec1 chloride/proton antiporter and find that these experimental photobleaching distributions predict the experimental labeling yields. A single data set can be used to make the prediction, but the accuracy is improved when we increase the amount of data included across multiple densities while considering the Poisson-like statistics of reconstitution. These fitted labeling yields are then used to predict the CLC-ec1 monomer signal, which is supported by the I201W/I422W experimental data. Finally, we estimate the dissociation constants of WT and I422W CLC-ec1 dimerization reactions, without prior information about fluorophore labeling yields. This work demonstrates an approach to measuring the fluorescent labeling yields directly from an experiment in real-time using a known dimeric control, representing a significant advance in the ability to determine fixed or dynamic stoichiometry for membrane proteins in membranes.

## Materials and methods

### The single-molecule photobleaching subunit capture approach

All experimental data presented here have been published previously [18, 25]. The analytical methods associated with this approach have been described in detail [11, 20] and are briefly outlined here. All data were collected on a construct of CLC-ec1 that contains two mutations to provide labeling specificity—C85A and H234C and the C-terminal hexahistidine-tag was left intact and is referred to as the 'WT' background. All subsequent mutations are constructed

upon the same background. 'RCLC' refers to two introduced mutations, previously found to enable spontaneous dimer cross-linking during the preparation of the protein–R230C and L249C [26], 'WT+Glut.' refers to 'WT' treated with glutaraldehyde, 'WW' refers to a known monomer sample, CLC-ec1 I201W/I422W [3] and 'W' refers to I422W [3, 25].

## Simulation of expected photobleaching probability distributions

In previous studies, we showed that the single-molecule photobleaching probability distributions measured using the subunit capture approach can be computationally predicted provided the experimental labeling yields, liposome size distribution, protein reconstitution yield, and monomer-dimer populations are known [18]. Thus, the single-molecule photobleaching probability distributions can be simulated using a MATLAB program that carries out the random process of subunit capture. This approach predicts changes in the photobleaching probability distribution associated with changes in liposome sizes and co-localization probabilities that agree with functional transport studies that also report on liposome occupancy [20]. The specific parameters in this model are listed in **Table 1.** The simulations yield the fraction of unoccupied vesicles, the fraction of vesicles that contain unlabelled protein, and the probability of vesicles that yield single ($P_1$), double ($P_2$) and more than double ($P_{3+}$) photobleaching steps as a function of the protein to lipid mole fraction. Thus, for a given liposome size distribution defined by $P_{radii}$, the expected ($P_1$, $P_2$, $P_{3+}$) vs. $\chi_{rec.}$ functions can be generated for any combination of $P_{fluor}$, $P_{bg}$ and $K_D$. In the case of a fixed dimer, one selects an arbitrarily strong dissociation constant, e.g. $K_D = 1 \times 10^{-100}$ subunits/lipid. Similarly, a fixed, unreactive monomer is simulated by selecting an arbitrarily weak dissociation constant, e.g. $K_D = 1 \times 10^{100}$ subunits/ lipid, yielding the expected photobleaching benchmarks for the experiments. For all of the modeling in this study, we used the 400 nm extruded 2:1 POPE/POPG outer radius distribution [25], considering all liposomes with radii > 20 nm can be occupied by dimers and a

**Table 1. Description of modeling parameters for protein reconstitution in liposomes.**

| Parameter | Description |
|---|---|
| **Fluorophore labeling:** | |
| $P_{bg}$ | the non-specific background labeling yield |
| $P_{fluor}$ | the overall fluorophore labeling yield per subunit, equal to $P_{bg} + P_{site}$, where $P_{site}$ is the labeling specific to the inserted cysteine site |
| **Protein Density:** | |
| $\chi_{rec.}$ | The reconstituted molar ratio of protein to lipids |
| $yield$ | The protein to lipid recovery yield after the reconstitution process |
| $\chi$ | The actual observed mole ratio, $\chi_{rec.} * yield$ |
| $\chi^*$ | The reactive observed mole ratio, $\chi/2$, accounting for random insertion of the subunits and assuming that the reaction only occurs between oriented subunits in the membrane |
| **Liposome size distribution:** | |
| $P_{radii}$ | The probability distribution of liposome sizes, usually experimentally measured by cryo-EM |
| $bias$ | Accounts for size biasing particularly for large proteins that cannot randomly incorporate into smaller liposomes. The bias excludes a certain number of small radii bins from the liposome size distribution. |
| **Monomer-Dimer equilibrium:** | |
| $K_D$ | Dissociation constant of the monomer-dimer reaction in the membrane, in units of subunit per lipid, i.e. $1/K_{eq}$. The fraction of dimer, $F_{Dimer}$, and corresponding number of monomer and dimer particles in the simulation is calculated based on the dimerization isotherm: $$F_{Dimer} = \frac{1 + \frac{4\chi^*}{K_D} - \sqrt{1 + \frac{8\chi^*}{K_D}}}{\frac{4\chi^*}{K_D}} \qquad (1)$$ |

reconstitution recovery yield = 0.5. A copy of all of the MATLAB programs used in this study is included as supplementary information.

## Estimation of $P_{bg}$ & $P_{fluor}$

When working with a known dimeric control, we fix our model to a dimer condition by setting the $K_D$ to an arbitrarily strong value such as 1 x 10$^{-100}$ subunits/lipid. With this, the fluorophore labeling yields, $P_{fluor}$ and $P_{bg}$, are iterated to determine which values provide the best fit for the known dimer experimental data, i.e. R230C/L249C CLC-ec1. The quality of the fit was quantified by calculating the sum of the squared residuals (SSR):

$$SSR(P_{fluor}, P_{bg}) = \sum_{i=1}^{N} \sum_{n=1,2,3+} (P_n^{expt} - P_n^{sim})_i^2 \qquad (2)$$

When iterating over a single parameter, the *SSR* data typically resembles a shallow parabola with a minimum *SSR* value. For peak determination, we apply an inverse transformation, which simply accentuates the maximum from the background:

$$SSR^{-1}(P_{fluor}, P_{bg}) = \frac{1}{SSR(P_{fluor}, P_{bg})} \qquad (3)$$

Using this distribution, we define the best fit-parameters $(P_{fluor}, P_{bg})_{max}$ corresponding to the maximum value of $SSR^{-1}$. The *SSR* values can be calculated for a single data point, a titration over different mole fraction densities or pooled over multiple samples. Since the magnitude of *SSR* depends on the number of data points included, we convert the *SSR* distribution to a probability distribution by carrying out a baseline correction of *(1-SSR)*, followed by an area normalization:

$$P_{SSR}(P_{fluor}, P_{bg}) = \frac{(1 - SSR) - min(1 - SSR)}{\sum_{P_{fluor}, P_{bg}} ((1 - SSR) - min(1 - SSR))} \qquad (4)$$

The values of $P_{SSR}$ reflect a discrete probability distribution over the user defined $(P_{fluor}, P_{bg})$ parameter space, where areas of higher probability are in better agreement with the experimental data than areas of lower probability. Note, the $P_{SSR}$ distribution is dependent on the limits of the parameters, and so we consistently set our search over the experimental limits of $P_{fluor}$ = {0.5, 1.0}, $P_{bg}$ = {0, 0.3}, resulting in 416 pairs of $(P_{fluor}, P_{bg})$. To calculate the uncertainty on $(P_{fluor}, P_{bg})_{max}$, we used a bootstrapping approach to obtain a $P_{SSR}$ probability weighted list of $(P_{fluor}, P_{bg})$ and calculated the standard deviation, $\sigma$, of the sampled population from the mode of the bootstrapped distribution $(x_{max})$ for each parameter:

$$\sigma = \sqrt{\frac{\sum (x_i - x_{max})^2}{N}} \qquad (5)$$

where $x_i$ represents each value in the list of the resampled $P_{fluor}$ or $P_{bg}$, $x_{max}$ is the value corresponding to the mode of $P_{fluor}$ or $P_{bg}$, and $N$ is the size of the bootstrapping selection, typically set to 10$^7$. For these calculations, the sampled populations are invariant for $N > 1000$.

## Estimation of $K_D$

In our previous analyses, an estimate of the $F_{Dimer}$ value corresponding to the experimental $(P_n^{expt}) = (P_1, P_2, P_{3+})$ was determined by least-squares estimation through a weighted average

of the expected monomer ($P_n^M$) and dimer ($P_n^D$) photobleaching distributions:

$$SSR(F_{Dimer}, P^M, P^D) = \sum_{n=1,2,3+} \left( P_n^{expt} - \left( (1 - F_{Dimer}) \cdot P_n^M + F_{Dimer} \cdot P_n^D \right) \right)^2 \tag{6}$$

This way of estimating $F_{Dimer}$ values assumes that liposomes containing monomers and liposomes containing dimers are two mutually exclusive populations. However, this is not technically correct as there is a small population of liposomes that contain both monomers and dimers. While this difference is small, the new approach presents a direct method of estimating the $K_D$ by iterative fitting of the experimentally determined $(P_1, P_2, P_{3+})$ vs. $\chi_{rec.}$ photobleaching data. To estimate the uncertainty of the fit, we followed the same bootstrapping approach used to estimate $P_{fluor}$ and $P_{bg}$ above. Here, the maximum $P_{SSR_i}$ value yields the best-fit estimate $K_{D,max}$ and bootstrapping yields $\sigma$.

## Statistical analyses

The parameter estimation of $P_{fluor}$, $P_{bg}$ or $K_D$ described above allows us to identify best-fit values to the photobleaching probability distributions, and standard deviations of the estimates from the bootstrapping analysis. In general, the estimation uncertainties are large because the $P_{SSR}$ distributions are shallow, which we interpret is due to the limited $(P_1, P_2, P_{3+})$ space. Still, we find that the best-fit values agree with experimental values, as is demonstrated by the agreement of $P_{fluor,fit}$ and $P_{bg,fit}$, and independent of large $\sigma$. However, for most studies, it is the sample variability that is of interest, not the estimate uncertainty. For this, we carry out independent parameter estimations from single samples, and then calculate the mean ± standard error (SE) over the sample set. To determine statistical differences between samples, we carry out a non-parametric two-tailed student's t-test on the different sets of sample data.

To ascertain whether two photobleaching probability distributions are statistically different, we carry out a chi-squared ($\chi^2$) analysis computed as follows:

$$\chi^2 = \sum \frac{(O_i - E_i)^2}{E_i} = \sum \frac{(P_n^M - P_n^D)^2}{P_n^D} \tag{7}$$

where the expected distribution is set as the dimer photobleaching probability distribution ($P_n^D$) and the observed distribution set to the monomer distribution ($P_n^M$). The null hypothesis ($H = 0$) is that the observed, monomer data are sampled from an expected population with dimer frequencies. We calculate the probability, $p$, of observing a discrepancy between monomer and dimer distributions. The test returns a rejection of the null hypothesis, $H = 1$, when $p \leq$ alpha and $H = 0$ for $p >$ alpha, indicating the observed distribution is statistically derived from the expected distribution. Note, we select alpha = 0.001, which provides reliable null hypothesis testing that allows discrimination between monomer and dimer distributions while accounting for our expected experimental sample variability. Finally, the difference between photobleaching probability distributions is quantified as the maximum scalar difference between monomer and dimer model $(P_1, P_2)$ distributions ($R_{max}$) as described previously [20]:

$$R_{max} = \sqrt{(P_1^M - P_1^D)^2 + (P_2^M - P_2^D)^2} \tag{8}$$

All statistical analyses are calculated in MATLAB and GraphPad Prism.

## Results

### The photobleaching probability distribution of CLC-ec1 dimers determines the fluorophore labeling yield per subunit

The CLC-ec1 chloride/proton antiporter participates in a reversible dimerization reaction within the lipid bilayer, meaning that the population dynamically shifts from monomers to dimers as a function of the protein density [18]. To measure this equilibrium constant, we relied on the subunit capture approach involving single-molecule photobleaching analysis. In order to quantify monomer and dimer populations from this analysis, we require an accurate measurement of the fluorophore labeling yield per subunit. Previously, we showed that CLC-ec1 labeling by Cy5-maleimide follows a two-site model (**Fig 1A**). One site represents non-specific background labeling, which can be measured for CLC-ec1 with the C85A construct that removes the partially accessible native cysteine. This construct yields $P_{bg}$ = 0.05–0.12 background labeling by Cy5-maleimide, as measured by UV-VIS spectrometry, presumably via the rare modification of primary amine groups on the protein. However, addition of the second labeling site, an aqueous accessible H234C provides rapid conjugation increasing the overall labeling yield $P_{fluor} = P_{site} + P_{bg} \approx 0.70$. In a single-molecule photobleaching analysis

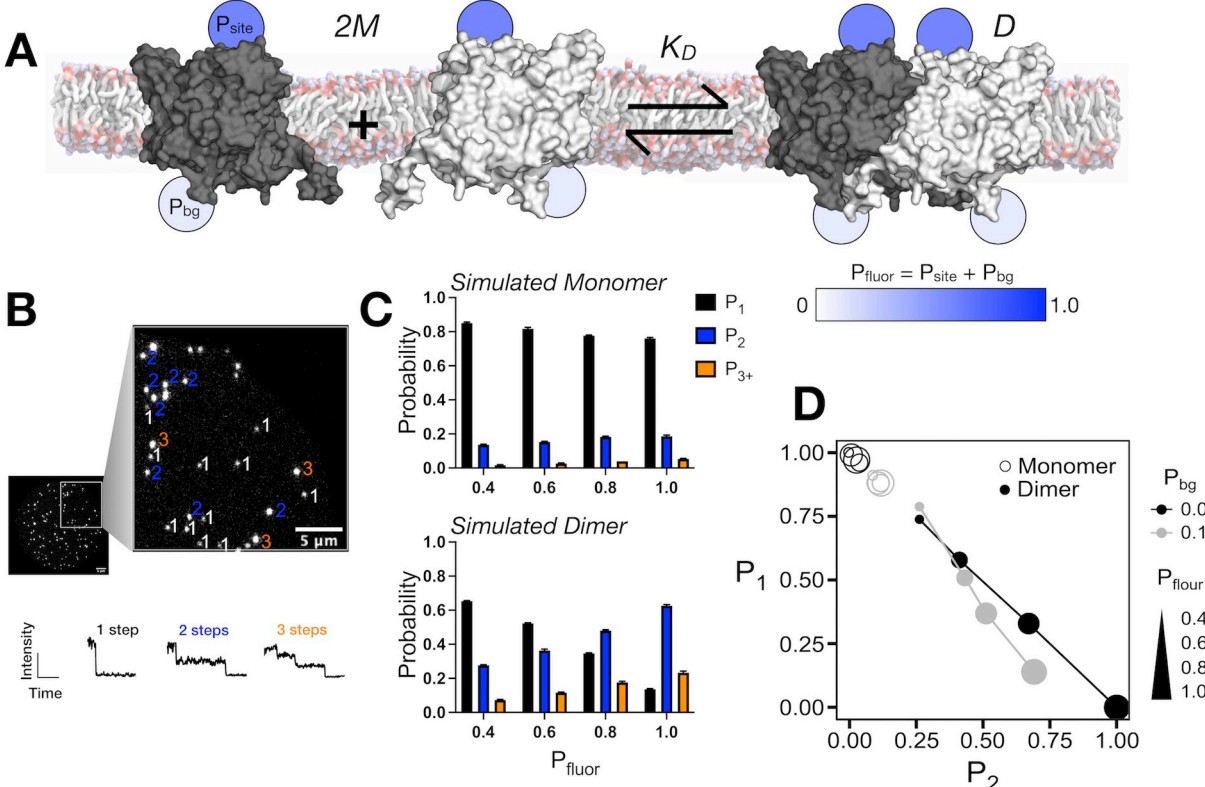

**Fig 1. Expected photobleaching probabilities for monomers and dimers at the single-molecule limit.** (A) The equilibrium dimerization reaction of two CLC-ec1 monomers (*2M*) forming a dimer (*D*) in the cellular membrane. In the labeling model, we consider that there is a high probability of labeling at an exposed cysteine site ($P_{site}$) and a lower probability of non-specific labeling contributing to the background ($P_{bg}$), which is measured in the protein sample without the cysteine. Experimentally the total labeling yield per subunit is measured, $P_{fluor} = P_{site} + P_{bg}$ for each protein sample. (B) Photobleaching of the fluorophores attached to the protein happens in a step wise fashion when imaged using a TIRF setup. (C) The monomeric and dimeric photobleaching probability distributions for labeling yields with $P_{bg}$ = 0.1, and total $P_{fluor}$ varied from 0.4 to 1.0, at a single-molecule density of $\chi_{rec.}$ = 1 x $10^{-6}$ subunits/lipid, i.e. << 1 protein species per liposome. Data reported as mean ± SE, for n = 3 simulation replicates. (D) Plot of $P_1$ vs. $P_2$ photobleaching probabilities showing the dependency of the dimer signal on the labeling yield and overall dynamic range between dimer and monomers.

experiment (**Fig 1B**) the number of fluorophores associated with a protein or liposome can be imaged using a total internal reflection fluorescence microscope and photobleaching steps can be counted directly by following the number of irreversible and instantaneous decreases in fluorescence intensity. In the single-molecule limit, the binomial labeling probability predicts that the monomer signal will be invariant with increasing labeling yield (**Fig 1C and 1D**). However, the expected single and double step photobleaching probabilities for a dimeric species provide a nearly linear correlation with labeling yield. This indicates that by imaging a dimeric control under sufficiently dilute conditions, labeling yields can be determined directly from the experimental photobleaching probability distribution.

To test this, we examined the experimental single-molecule photobleaching probability distributions for the cysteine cross-linked CLC-ec1 construct, R230C/L249C (RCLC) collected previously [25]. This construct has been demonstrated to be dimeric in membranes while maintaining chloride/proton transport function [26]. In our samples, the protein is in liposomes, therefore, the single-molecule limit depends on the Poisson-like probability distribution of liposome occupancies. This distribution can be modeled mathematically provided the subunit/lipid mole fraction, reaction $K_D$, liposome size distribution and fluorescent labeling yields are all known [20]. For these samples, the experimental labeling yields were measured as $(P_{fluor}, P_{bg})_{expt.}$ = $(0.72 \pm 0.05, 0.11 \pm 0.01)$, $n = 3–5$. We tested the predictive power of a single reconstitution density by carrying out an iterative search of the Poisson simulation as a function of $P_{fluor}$ and $P_{bg}$ while keeping the $K_D$ constant to a value favoring dimerization, i.e. $K_D = 1 \times 10^{-100}$ subunits/lipid. Comparing the results for several reconstitution densities of the same sample (**Fig 2A–2C**), we observe that the labeling yield is consistently predicted, with the mean ± standard deviation ($\sigma$) of labeling over the first four densities $(P_{fluor}, P_{bg})_{\chi 1–4,max}$ = $(0.72 \pm 0.03, 0.05 \pm 0.04)$ (**Fig 2D and Table 2**). However, the prediction deviates at the highest density measured, $\chi_{rec.} = 2 \times 10^{-5}$ subunits/lipid $(P_{fluor}, P_{bg})_{\chi 5,max}$ = $(0.50, 0.02)$. This is the density where liposomes become occupied by multiple protein copies and indicates that this over-filling obscures the predictive power of this approach. Therefore, we only use this approach for single-molecule densities, and for CLC-ec1 dimers this pertains to where $\chi_{rec.} < 2 \times 10^{-5}$ subunits/lipid.

The fitting estimate can be refined further by including multiple data sets collected in this single-molecule range. We did this by pooling all densities for a single sample titration where $\chi_{rec.} < 2 \times 10^{-5}$ subunits/lipid (**Fig 3A and Table 3**). Because each sample was prepared separately, they have different labeling yields, yet the parameter fitting gives estimates that align with the mean experimental labeling values (**Fig 3B**). Furthermore, global fitting is achieved by pooling all of the data for $\chi_{rec.} < 2 \times 10^{-5}$ subunits, from all of the samples studied (**Fig 3C**). This overall fit predicts a labeling yield of $(P_{fluor}, P_{bg})_{max}$ = $(0.72, 0.10)$, $(P_{fluor}, P_{bg})_{boot.}$ = $(0.66 \pm 0.17, 0.24 \pm 0.13)$ (**Table 4**) that corresponds to the experimental average. As a final test, we simulated the photobleaching probability distributions using the estimated labeling yields from the global fitting, revealing how the model reflects the trend and quantities of the experimental data (**Fig 3D**). Therefore, these results demonstrate that proteoliposomes of a fixed dimer species can be used to accurately estimate the labeling yield of the sample and experimental photobleaching probability distribution when the protein is sufficiently diluted in the membrane.

## Estimating monomers, dimers and dimerization reaction equilibria from photobleaching probability distributions without prior knowledge of fluorophore labeling

Next, we tested whether the estimated $(P_{fluor}, P_{bg})_{max}$ and $(P_{fluor}, P_{bg})_{boot.}$ Determined from the RCLC dimer control could be used in quantifying the reactive oligomerization of other CLC-ec1 constructs. Using the global labeling estimates (**Fig 3C and Table 4**) and an iterative

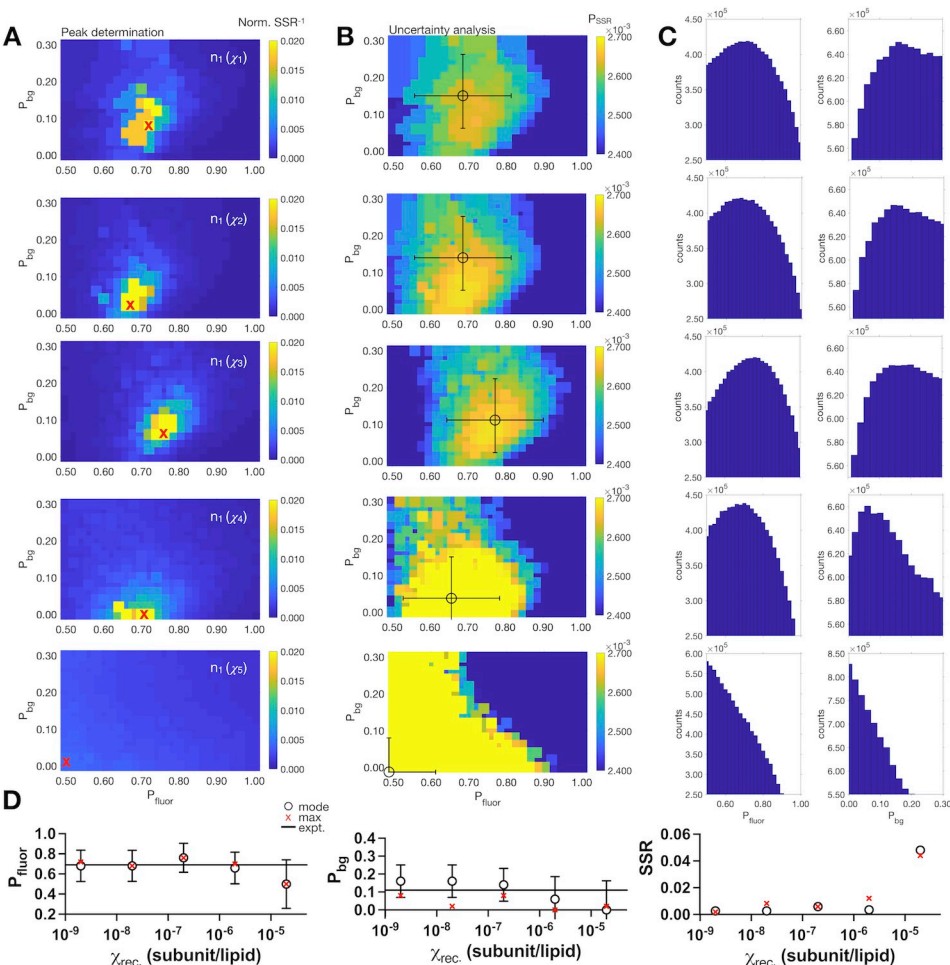

**Fig 2. Single-molecule photobleaching estimation of ($P_{fluor}$, $P_{bg}$) for covalently cross-linked CLC-ec1 dimers across a wide range of protein densities.** (A) Heatmaps of the inverse normalized sum of squared residuals (*Norm. SSR^{-1}*) over the fluorophore labeling parameter space of ($P_{fluor}$, $P_{bg}$). The experimental photobleaching data used in this analysis are from [25] for the covalently cross-linked R230C/L249C CLC-ec1 dimer, for a single sample ($n_1$) across a 5-magnitude range in mole fraction densities: $\chi_1 = 2 \times 10^{-9}$, $\chi_2 = 2 \times 10^{-8}$, $\chi_3 = 2 \times 10^{-7}$, $\chi_4 = 2 \times 10^{-6}$, $\chi_5 = 2 \times 10^{-5}$ subunits/lipid. Maximum value, corresponding to the ($P_{fluor}$, $P_{bg}$) pair that best fits the experimental data is indicated by the red "x". (B) Heatmaps of the probability distribution of the sum of squared residuals, $P_{SSR}$. (C) Bootstrapping analysis from the $P_{SSR}$ distribution. The sampling number, $N$, is set to $10^7$. The mode and standard deviation around the mode, $\sigma$, from each bootstrapped distribution are marked in panel (B) with the circle and error bars, respectively. (D) Mode $\pm \sigma$ (circle $\pm$ error bars) and max values (red "x") of $P_{fluor}$ and $P_{bg}$ from the bootstrapping analysis along with SSR values compared to the experimental data.

**Table 2. Fitted vs. experimental parameters for RCLC sample 1.**

| $\chi_{rec.}$ (subunits/lipid) | ($P_{fluor}$, $P_{bg}$)$_{max}$ | $SSR_{expt.}$ | ($P_{fluor} \pm \sigma$, $P_{bg} \pm \sigma$)$_{boot.}$ | $SSR_{expt.}$ |
|---|---|---|---|---|
| $2 \times 10^{-9}$ | (0.72, 0.08) | 0.0018 | (0.68 ± 0.16, 0.16 ± 0.09) | 0.0026 |
| $2 \times 10^{-8}$ | (0.68, 0.02) | 0.0082 | (0.68 ± 0.16, 0.16 ± 0.09 | 0.0026 |
| $2 \times 10^{-7}$ | (0.76, 0.08) | 0.0058 | (0.76 ± 0.15, 0.14 ± 0.09) | 0.0058 |
| $2 \times 10^{-6}$ | (0.70, 0.00) | 0.012 | (0.66 ± 0.16, 0.06 ± 0.13) | 0.0034 |
| $2 \times 10^{-5}$ | (0.50, 0.02) | 0.044 | (0.50 ± 0.24, 0.00 ± 0.16) | 0.048 |

($P_{fluor}$, $P_{bg}$)$_{expt.}$ = (0.69, 0.11) and $SSR_{expt.} = (P_{fluor,expt.}-P_{fluor,fit})^2 + (P_{bg,expt.}-P_{bg,fit})^2$, where "*fit*" represents the best-fit (*max*) or bootstrapped (*boot.*) parameters.

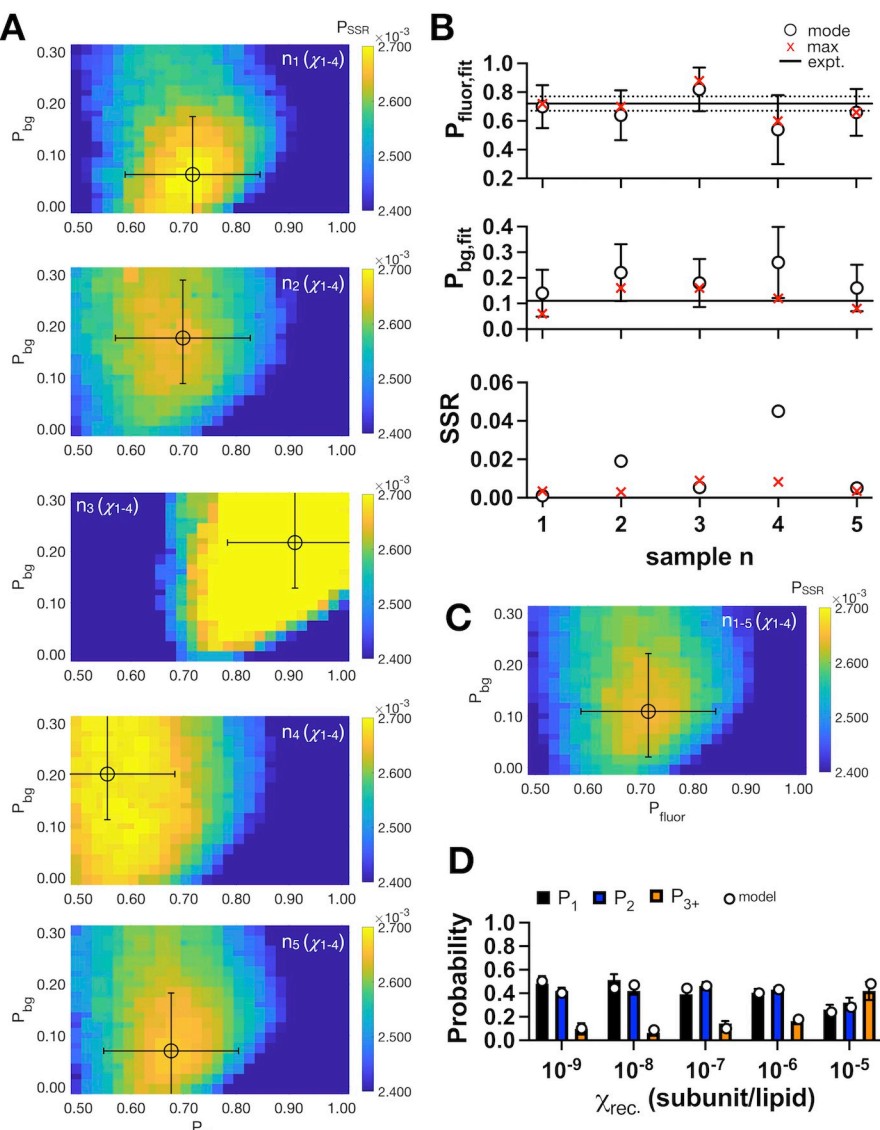

**Fig 3. Single-molecule photobleaching estimation of fluorophore labeling recapitulates sample variability.** (A) Heatmap of $P_{SSR}$ over the parameter space of ($P_{fluor}$, $P_{bg}$) for different experimental samples, $n$, of the covalently cross-linked R230C/L249C CLC-ec1 dimer. The circle and error bars reflect the mode $\pm$ $\sigma$ from the bootstrapping analysis. (B) The maximum and mode values of $P_{fluor}$ and $P_{bg}$, along with $SSR$ compared to the experimental values, dotted line. (C) Global fit of ($P_{fluor}$, $P_{bg}$) obtained from pooling experimental data for samples $n$ = 1–5. (D) Experimental photobleaching data ($P_1$, $P_2$, $P_{3+}$), along with the modeled probability distribution (circles) using ($P_{fluor}$, $P_{bg}$) = (0.72,0.10) corresponding to the maximum value from the global fit in panel (C).

search of $K_D$ using our Poisson model, we compared the ability to use RCLC photobleaching data as a proxy for the labeling yield compared to direct quantification of the labeling yields of the purified protein by UV-VIS previously obtained [18, 25]. First, we analyzed another dimer model, WT CLC-ec1 treated with the cross-linking agent glutaraldehyde (WT+ Glut., **Fig 4A**). While glutaraldehyde treatment yields an irreversible dimer, a significant loss of protein function was also observed indicating a change in protein structure accompanies crosslinking that does not occur in the disulfide linked RCLC [25]. The fits show similar $P_{SSR}$ for the experimental or RCLC derived labeling yields and becomes maximal as the search moves leftwards,

**Table 3. Fitted vs. experimental parameters for RCLC samples 1–5, and $\chi_{rec.} = 2 \times 10^{-9}$ to $2 \times 10^{-6}$.**

| Sample | $(P_{fluor}, P_{bg})_{expt.}$ | $(P_{fluor}, P_{bg})_{max}$ | $SSR_{expt.}$ | $(P_{fluor} \pm \sigma, P_{bg} \pm \sigma)_{boot.}$ | $SSR_{expt.}$ |
|---|---|---|---|---|---|
| 1 | (0.69, 0.11) | (0.72, 0.06) | 0.0034 | (0.70 ± 0.15, 0.14 ± 0.11) | 0.0010 |
| 2 | (0.72, 0.11) | (0.70, 0.16) | 0.0029 | (0.64 ± 0.17, 0.22 ± 0.11) | 0.019 |
| 3 | (0.80, 0.11) | (0.88, 0.16) | 0.0089 | (0.82 ± 0.15, 0.18 ± 0.09) | 0.0053 |
| 4 | (0.69, 0.11) | (0.60, 0.12) | 0.0082 | (0.54 ± 0.24, 0.26 ± 0.14) | 0.045 |
| 5 | (0.71, 0.11) | (0.66, 0.08) | 0.0034 | (0.66 ± 0.16, 0.16 ± 0.09) | 0.0050 |

Experimental labeling values (*expt.*) from [25].

towards more stable $K_D$ values, in line with an irreversible dimer. The plateau in $P_{SSR}$ indicates that all $K_D$ values beyond the lower limit provide the same fit, indicating that a single $K_D$ cannot be estimated. Similarly, the analysis of a known monomer sample, CLC-ec1 I201W/I422W, 'WW' [3, 25] shows nearly identical fits for $(P_{fluor}, P_{bg})_{max}$ and $(P_{fluor}, P_{bg})_{boot.}$ compared to the experimental labeling yields. The WW samples show a maximal $P_{SSR}$ as the $K_D$ approaches higher values, indicating that there is no appearance of a dimerization reaction in this sample and an inability to predict a single $K_D$.

Finally, we analyzed two known reactive species, 'WT' CLC-ec1 and I422W CLC-ec1 I422W, 'W'. The analysis shows the $P_{SSR}$ trends for the best-fit and bootstrapped labeling yields are comparable to the experimental yields, but this analysis shows a discrete peak over a specific range of $K_D$ *values*, specific to each construct. For WT, we previously estimated $K_{D,WT} = 5.4 \times 10^{-8}$ subunit/lipid [25] by determining $F_{Dimer}$ from a series of photobleaching probability distributions at different $\chi^*$ values, using a least-squares estimate to the weighted average of monomer and dimer control distributions. The $F_{Dimer}$ vs. $\chi^*$ data is then fit to a dimerization equilibrium isotherm to determine $K_D$. This way of estimating $F_{Dimer}$ values assumes that liposomes containing monomers and liposomes containing dimers are two mutually exclusive populations. However, this is not technically correct as there is a small population of liposomes that contain both monomers and dimers. The iterative fitting approach presented here (**Fig 4A**) provides a direct method of estimating the $K_D$ to the experimentally determined $(P_1, P_2, P_{3+})$ vs. $\chi_{rec.}$ photobleaching data, and allows the consideration that monomer and dimer species can occupy the same liposome. With this, we obtain best-fit $K_D$ values of $1.5 \times 10^{-9}$, $8.0 \times 10^{-10}$ and $1.5 \times 10^{-9}$ subunits/lipid for the experimental, and RCLC max and bootstrap estimated labeling yields, respectively (**Table 5**). In addition, W, a destabilized CLC-ec1 dimer was previously reported to have a $K_{D,W} = 6.7 \times 10^{-7}$ subunits/lipid based on the weighted least-squares method. The new approach yields $K_{D,W}$ of $1.5 \times 10^{-7}$, $9 \times 10^{-8}$ and $7.0 \times 10^{-8}$ subunits/lipid for the experimental, and RCLC determined labeling yields, respectively. The estimated $K_Ds$ using $(P_{fluor}, P_{bg})_{max}$ provide $(P_1, P_2, P_{3+})$ distributions that correspond well to the experimental photobleaching probability distributions (**Fig 4B**). The new approach yields $K_D$ values that are about one magnitude shifted, but the difference between WT and W dissociation constants is maintained across both methods (**Fig 4C**). There is complete overlap of the $K_D$ estimates based on the experimental labeling yields, or those determined from fitting of the RCLC distributions,

**Table 4. Global fitting over all RCLC samples 1–5, and $\chi_{rec.} = 2 \times 10^{-9}$ to $2 \times 10^{-6}$.**

| $(P_{fluor} \pm std, P_{bg} \pm std)_{expt.}$ | $(P_{fluor}, P_{bg})_{max}$ | $SSR_{expt.}$ | $(P_{fluor} \pm \sigma, P_{bg} \pm \sigma)_{boot}$ | $SSR_{expt.}$ |
|---|---|---|---|---|
| (0.72 ± 0.05, 0.11 ± 0.01) | (0.72, 0.10) | 0.00010 | (0.66 ± 0.17, 0.24 ± 0.13) | 0.021 |

Experimental labeling values (*expt.*) reported as mean ± standard deviation (*std*), $n$ = 5 samples [25].

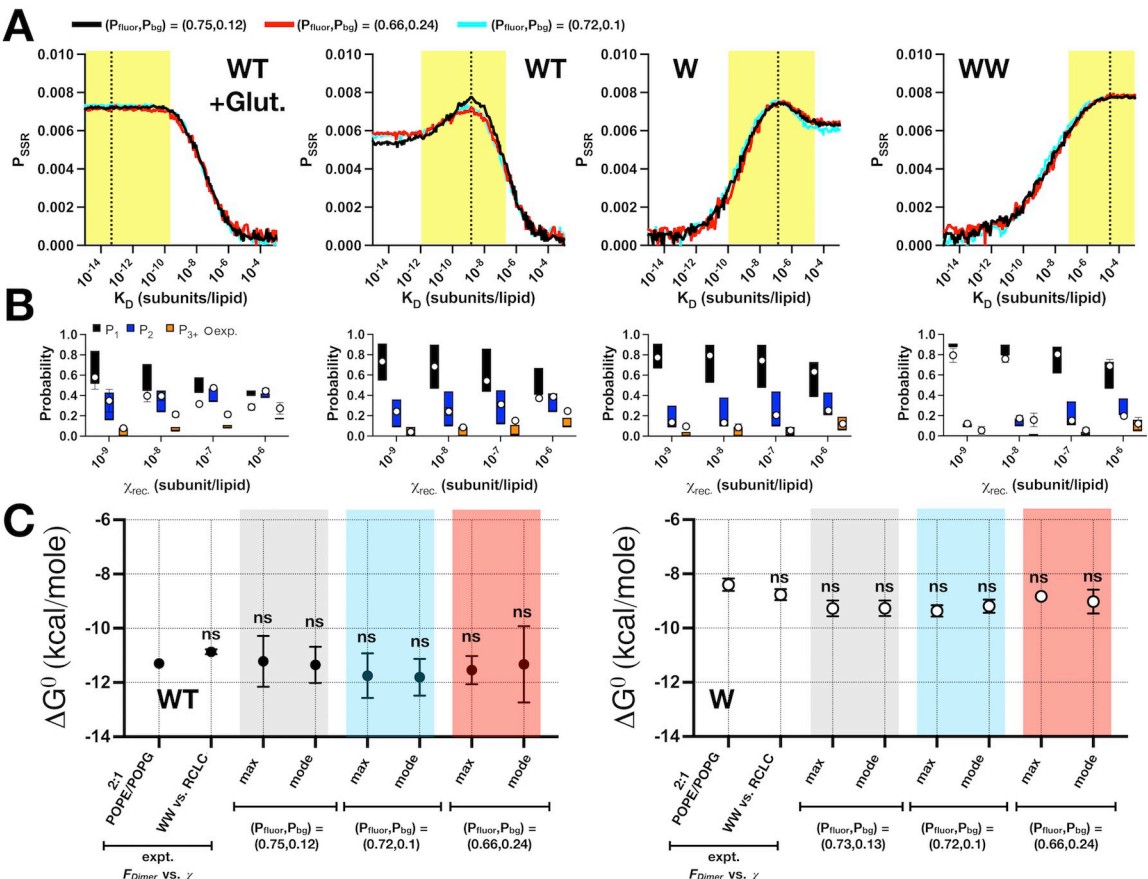

**Fig 4. Estimating $K_D$ values for CLC-ec1 dimerization in membranes.** (A) $P_{SSR}$ for dimeric glutaraldehyde cross-linked WT, reactive WT, reactive I422W, 'W', and monomeric I201W/I422W, 'WW', as a function of the dissociation constant parameter $K_D$. Curves reflect the model using the experimental labeling parameters $(P_{fluor}, P_{bg})_{expt.}$—black or the R230C/L249C fitted labeling parameters $(P_{fluor}, P_{bg})_{max}$—cyan, $(P_{fluor}, P_{bg})_{boot.}$—red. Dotted line represents the maximum $P_{SSR}$ value and best-fit $K_D$ using the $(P_{fluor}, P_{bg})_{max}$ labeling parameters, and the yellow box reflects the uncertainty based on the bootstrapping analysis. (B) Range of photobleaching probability distribution ($P_1$, $P_2$, $P_{3+}$) simulated using $(P_{fluor}, P_{bg})_{max}$ and $K_{D,boot.}$—$\sigma$, $K_{D,boot.}$, $K_{D,boot.}$ + $\sigma$ and agreement with experimental data (white circles). (C) $\Delta G^0$ for WT and W [20] based on least-squares estimation of $F_{Dimer}$ from expected monomer and dimer photobleaching probability distributions compared to the direct fitting of the photobleaching probability distribution while iterating over $K_D$ as a parameter. $\Delta G^0 = -RTln(K_{eq}\chi^\circ)$, where $K_{eq} = 1/K_D$ and $\chi^\circ = 1$ subunit/lipid represents the mole fraction standard state. Results shown for the best-fit, 'max', value as well as the mode of the bootstrapping analysis for $(P_{fluor}, P_{bg})_{expt.}$—grey, compared to the R230C/L249C fitted labeling parameters $(P_{fluor}, P_{bg})_{max}$—cyan, $(P_{fluor}, P_{bg})_{boot.}$—red. All data are shown as mean ± SE of fits of independent data sets and statistical significance is calculated via t-test.

showing that we can determine dynamic stoichiometry of CLC-ec1 by only having a dimeric control and no prior knowledge of the labeling yield. Further, by employing an iterative search of the Poisson simulation we allow for a more complete interrogation of the uncertainties of our $K_D$ estimates by employing a bootstrapping analysis on $P_{SSR}$. While the standard deviations remain quite large, they provide a description of the range of $K_D$s that can suitably fit to the experimental photobleaching data, emphasizing that this approach is most powerfully used when examining changes in dimerization behavior between protein populations.

## Defining an operational range for fluorophore labeling for single-molecule photobleaching studies of dimerization

As a final step in our investigation, we carried out a statistical analysis to determine an appropriate operational range for fluorophore labeling yields that can be used for estimating $K_D$s in

**Table 5. Direct determination of $K_D$s from photobleaching distributions.**

| Construct | method | $(P_{fluor.} \pm \sigma, P_{bg.} \pm \sigma)_{expt.}$ | $K_D^*$ (sub./lipid) | $K_{D,max.}$ (sub./lipid) | $(K_D - \sigma, K_D, K_D + \sigma)_{boot.}$ (sub./lipid) |
|---|---|---|---|---|---|
| WT + Glut. | expt. | $(0.70 \pm 0.04, 0.05)$ | ND | $(8.0 \times 10^{-11})^*$ | $(8.1 \times 10^{-19}, 1.0 \times 10^{-14}, 1.2 \times 10^{-10})\#$ |
| | max | $(0.72, 0.10)$ | | $(6.5 \times 10^{-14})\#$ | $(2.7 \times 10^{-14}, 9.0 \times 10^{-12}, 3.0 \times 10^{-9})\#$ |
| | boot. | $(0.66 \pm 0.17, 0.24 \pm 0.13)$ | | $(8.5 \times 10^{-14})\#$ | $(9.4 \times 10^{-16}, 9.5 \times 10^{-13}, 9.6 \times 10^{-10})\#$ |
| WT | expt. | $(0.75 \pm 0.04, 0.12)$ | **$5.4 \times 10^{-9}$** $1.1 \times 10^{-8}$ | $1.5 \times 10^{-9}$ | $(1.2 \times 10^{-12}, 8.5 \times 10^{-10}, 6.3 \times 10^{-7})$ |
| | max | $(0.72, 0.10)$ | | $8.0 \times 10^{-10}$ | $(9.7 \times 10^{-13}, 9.0 \times 10^{-10}, 8.3 \times 10^{-7})$ |
| | boot. | $(0.66 \pm 0.17, 0.24 \pm 0.13)$ | | $1.5 \times 10^{-9}$ | $(4.0 \times 10^{-12}, 2.5 \times 10^{-10}, 1.6 \times 10^{-7})$ |
| W | expt. | $(0.73 \pm 0.03, 0.13 \pm 0.03)$ | **$6.7 \times 10^{-7}$** $3.4 \times 10^{-7}$ | $1.5 \times 10^{-7}$ | $(9.9 \times 10^{-11}, 7.0 \times 10^{-8}, 5.0 \times 10^{-5})$ |
| | max | $(0.72, 0.10)$ | | $9.0 \times 10^{-8}$ | $(1.0 \times 10^{-10}, 7.0 \times 10^{-8}, 4.9 \times 10^{-5})$ |
| | boot. | $(0.66 \pm 0.17, 0.24 \pm 0.13)$ | | $3.0 \times 10^{-7}$ | $(1.2 \times 10^{-10}, 9.0 \times 10^{-8}, 6.7 \times 10^{-5})$ |
| WW | expt. | $(0.68 \pm 0.06, 0.16 \pm 0.01)$ | ND | $(2.5 \times 10^{-3})\#$ | $(1.5 \times 10^{-7}, 3.0 \times 10^{-3}, 6.0 \times 10)\#$ |
| | max | $(0.72, 0.10)$ | | $(7.0 \times 10^{-5})\#$ | $(1.1 \times 10^{-8}, 5.0 \times 10^{-6}, 2.3 \times 10^{-3})\#$ |
| | boot. | $(0.66 \pm 0.17, 0.24 \pm 0.13)$ | | $(4.5 \times 10^{-5})\#$ | $(1.5 \times 10^{-7}, 9.0 \times 10^{-3}, 5.2 \times 10^{2})\#$ |

*, indicates prior fitting results from least-squares estimation of $F_{Dimer}$, with the bolded values using monomer and dimer distributions based on modeling of the 2:1 POPE/POPG 400 nm extruded outer membrane liposome size distribution [18], and the other value using the experimental I201W/I422W and R230C/L249C distributions [25]. #, indicates fitted $K_D$ values, but do not support equilibrium reactions due to insufficient data.

a dimerization reaction using the single-molecule subunit capture photobleaching method. To determine this, we generated $(P_1, P_2, P_{3+})$ photobleaching probability distributions for monomer and dimer models as a function of total labeling yield, $P_{fluor}$, background labeling yield, $P_{bg}$ while setting $P_{bg} < P_{site}$ and the mole fraction density, $\chi$, examining the range of parameters where the distributions have the ability to discriminate between populations. The $\chi^2$ values between the monomer and dimer distributions increases as $P_{fluor}$ increases, and decreases as the protein density increases (**Fig 5A**). However, significance testing at a probability limit of alpha = 0.001 indicates that for $\chi \leq 10^{-6}$ subunits/lipid labeling yields as low as $P_{fluor} = 0.4$ are capable of discriminating between monomer and dimer populations, assuming $P_{bg} \leq 0.1$ (**Fig 5B**). Comparing how the monomer vs. dimer signal shifts in $(P_1, P_2)$ space shows how the signal remains well separated at lower labeling yields at the lower density of $\chi = 10^{-6}$ subunits/lipid, even when considering experimental variability in the data (**Fig 5C**). However, at $\chi = 10^{-5}$ subunits/lipid, where liposomes begin to become saturated with protein, the signals are overlapping at lower labeling conditions, and there is no ability to discriminate between the populations. Thus, considering both experimental variability in the photobleaching data, and the need to have a multiple density points that span the $K_D$ of the reaction, we suggest an operational fluorophore labeling condition of $P_{fluor} \geq 0.7$ and $P_{bg} \leq 0.1$ if density points $\chi > 10^{-6}$ subunits/lipid are to be included. However, if the reaction is observable at $\chi \leq 10^{-6}$ subunits/lipid, then labeling conditions of $P_{fluor} \geq 0.4$, $P_{bg} \leq 0.1$ will offer an ability to discriminate between monomers and dimers. However, it should be noted that the robustness in the ability to discriminate between populations increases substantially as $P_{fluor}$ increases to 1, and $P_{bg}$ decreases to 0. Therefore we recommended optimizing the experimental labeling yields as close to this condition as is possible for any new sample that is being studied by the single-molecule photobleaching subunit capture approach.

## Discussion

In this study, we demonstrate that the single-molecule photobleaching probability distribution of a known dimeric control of CLC-ec1, R230C/L249C, provides an accurate prediction of the fluorophore labeling yield per subunit. Furthermore, given that the labeling reaction is similar

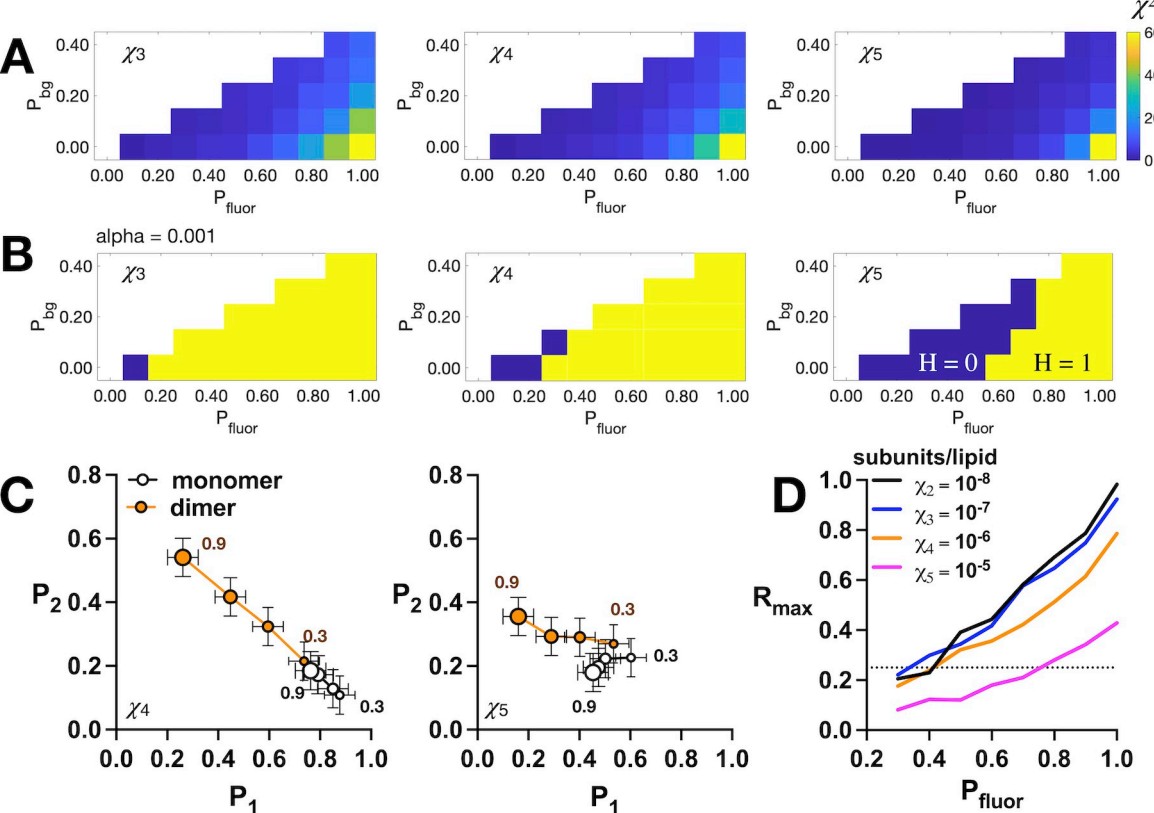

**Fig 5. The operational range for fluorophore labeling yields.** (A) Chi-squared ($\chi^2$) analysis between monomer and dimer model photobleaching probability distributions over ($P_{fluor}$, $P_{bg}$) fluorophore labeling parameter space for $P_{bg} < P_{site} = P_{fluor} - P_{bg}$. $\chi^2$ heatmaps are shown for mole fraction densities $\chi_3 = 2 \times 10^{-7}$, $\chi_4 = 2 \times 10^{-6}$, $\chi_5 = 2 \times 10^{-5}$ subunits/lipid. (B) Heatmaps of null hypothesis testing for the $\chi^2$ values in (A) for alpha = 0.001 significance. The null hypothesis is that the monomer and dimer distributions are the same, where H = 1 (yellow) indicates a rejection of the null hypothesis, and H = 0 (purple) indicates that the null hypothesis cannot be rejected at the indicated significance level. (C) $P_1$ vs. $P_2$ for monomer (white) and dimer (orange) models for $P_{fluor} = 0.3$ to 0.9 and $P_{bg} = 0.1$. Symbol size increases with increasing $P_{fluor}$ with endpoints labelled as shown. Error bars depict representative standard deviation values of the experimental data for RCLC, std = ± 0.06. (D) Maximal scalar distance, $R_{max}$, between ($P_1$, $P_2$) signals from monomer and dimer model distributions as a function of mole fraction density. The background labeling yield is set to $P_{bg} = 0.1$. The dotted line indicates $R_{max} = 0.25$ that corresponds to the significance testing cutoff in the $\chi^2$ analysis in (B).

to other CLC-ec1 constructs, this provides a means for accurate estimation of stoichiometry and in-membrane reactive $K_D$ values for a range of other CLC-ec1 reactions without the need to measure the fluorophore labeling yield *a priori* on the purified protein. This work reflects a significant advance in the ability to accurately quantify dynamic reactions of membrane protein oligomerization in membranes.

In order to quantify membrane protein stoichiometry from single-molecule photobleaching data, the probability that an observable fluorophore is conjugated to a protein subunit must be known. When working with fluorescent protein tags, such as GFP, this means that the maturation yield must be known in the context of the experimental environment [10, 21]. As previously pointed out, without this critical information, the oligomeric state of the tagged protein remains an ill-posed inference problem [24]. On the other hand, for purified proteins the fluorophore labeling yield may be quantified in bulk using absorbance spectroscopy, comparing the ratio of molecular amounts estimated from the absorbance of the fluorophore and protein while correcting for spectral overlap. While direct, this approach requires large amounts of protein. It also relies on the assumption that there is no change in the fluorescence of the label from the

point of protein purification to the measurement of the sample. However, a membrane protein may go through many different steps following the initial quantification in detergent. For example, in our studies, fluorescently labelled CLC-ec1 subunits are reconstituted into lipid bilayers in a multi-day dialysis procedure. Afterwards, the proteoliposomes undergo multiple freeze/thaw cycles to form multi-lamellar vesicles, and then these membranes are then incubated in dark conditions at the desired temperature and required incubation time, which can extend anywhere from 3 days to months. The photostability of the fluorophore under such extreme conditions must always be questioned. Over time, the environment of the fluorophore could change due to oxidation, protein unfolding, or other reasons [27]. Therefore, it is best to have a direct approach to quantify the labeling yield at the time and under the conditions that the data is collected. With our results here, we demonstrate that this can be readily done by preparing a disulfide cross-linked form of the protein of interest and using it as a method for determining the fluorophore labeling yield directly from the photobleaching probability distribution, which can be obtained alongside the protein of interest.

Another benefit of this approach is that it reduces the requirement for bulk-level protein purification. In principle, one could blindly label a small amount of protein and reconstitute it into liposomes and follow through with single-molecule studies. The protein quantity does not need to be known at the reconstitution step, but can be determined from the reconstituted samples in a method similar to how we determine our observed mole fraction. Here, we measure the actual protein and lipid recovery of our higher density samples $\chi_{rec.} > 10^{-8}$ to $10^{-5}$ subunits/lipid directly, by phosphate quantification of the moles of lipids and moles of fluorescently labelled protein in a fluorescence calibration assay [18, 28]. Thus, the total amount of fluorophore in the protein sample can be measured directly, and determining the subunit labeling yield from the photobleaching distribution of a parallel dimer control sample will give the total protein density pertaining to the reaction. This method could also be applied to fluorescent protein fusions such as GFP, where the actual GFP maturation fraction must be inferred from the experimental data [24]. Previous attempts have tried to mitigate these problems by fitting the data to a binomial probability distribution and deducing the oligomeric state from the highest observed step count [10]. However, this method becomes complicated if there are multiple oligomeric species in the system. Here, we suggest using a known dimer control that does not engage in higher-order oligomerization will offer a direct approach the determining the subunit labeling yield in intact membranes.

An important caveat is that this method assumes that the labeling behavior for a certain protein construct will be comparable to the dimer control. But, we can envision situations where this may not be the case. For instance, if a mutation stabilizes a particular state where the labeling site is no longer reactive. Alternatively, different solvent environments, e.g. detergent micelles vs. amphiphiles, may reduce the accessibility of the reactive group. If this occurs, then the decrease in labeling yield could incorrectly be interpreted as a decrease in oligomerization instead of a reduction in labeling. Thus, it is always advised to check the reactivity of the labeling site by the Ellman's assay, or carrying out mole-fraction quantification or selectively scale up some purifications to check the bulk labeling yields. Furthermore, it is recommended to test out multiple labeling sites to see if there is a dependency on the measurement of the reaction. For a protein like CLC-ec1 that undergoes minor conformational changes, we have found the labeling yields to be consistent across over 15 different constructs studied [11, 25, 29].

## Conclusion

In conclusion, we validate a method for single-molecule determination of fluorophore labeling yields based on a dimer-control photobleaching probability distributions. We tested this

approach on the CLC-ec1 homodimer in membranes and demonstrate the ability to discriminate monomer and dimer species, and accurately measure $K_Ds$ for reactive dimers, all without prior bulk quantification of the fluorophore labeling yields. This approach greatly facilitates single-molecule photobleaching experiments, and adds rigor in providing the ability for on the fly quantification of the fluorescent labeling yield directly from the experimental data. Our analysis indicates that in the range of $\chi_{rec.} = 10^{-9}$ to $10^{-5}$ subunits/lipid, fluorophore labeling yields of $P_{fluor} \geq 0.7$ and $P_{bg} \leq 0.1$ are capable of discriminating between monomer and dimer populations, whereas for $\chi_{rec.} \leq 10^{-6}$ subunits/lipid, lower labeling yields such as $P_{fluor} \geq 0.4$ can be tolerated. While we show that this analysis can be used for the complex example of membrane proteins in lipid bilayers, it can also be extended to single-molecule dilutions of soluble proteins or membrane proteins in detergent micelles for a broader use of the approach.

## Supporting information

**S1 File. MATLAB program PfluorPbgfitting.mlapp.** This program takes the liposome size distribution, the reconstitution yield, the smallest liposome size radius that allows for occupancy of the liposome by two subunits, the experimental dimer photobleaching distribution and the ($P_{fluor}$, $P_{bg}$) range as inputs. It then generates raw *SSR* values of the experimental data vs. the model while iterating over ($P_{fluor}$, $P_{bg}$) and further carries out the peak determination from *Norm. SSR$^{-1}$* and the variance analysis on bootstrapping of $P_{SSR}$. This MATLAB application was written using the Mac version of MATLAB (R2020b). Previous versions may not support the application and different operating systems might need adjustments to the code. The following toolbox needs to be installed for running the app: "Statistics and Machine Learning Toolbox". A detailed step-by-step instruction of using the app can be found at: https://github.com/tnozturk/smPBfit.
(MLAPP)

**S2 File. MATLAB program KDFitting.mlapp.** This program takes the liposome size distribution, the experimental photobleaching distribution, the designated ($P_{fluor}$, $P_{bg}$) values, the reconstitution yield, the smallest liposome size radius that allows for occupancy of the liposome by two subunits, and the desired $K_D$ range and step size as inputs. It then generates the raw *SSR* values of the experimental data vs. the model while iterating over $K_D$ and further carries out the peak determination from *Norm. SSR$^{-1}$* and the variance analysis on bootstrapping of $P_{SSR}$. This MATLAB application was designed using the Mac version of MATLAB (R2020b). Previous versions might not support the application and different operating systems might need adjustments to the code. The following toolbox needs to be installed for running the app: "Statistics and Machine Learning Toolbox". A detailed step-by-step instruction of using the app can be found at: https://github.com/tnozturk/smPBfit.
(MLAPP)

**S3 File. Source data file.**
(XLSX)

## Acknowledgments

We thank the members of the Robertson Lab for helpful discussions.

## Author Contributions

**Conceptualization:** Janice L. Robertson.

**Data curation:** Melanie Ernst, Tugba N. Ozturk, Janice L. Robertson.

**Formal analysis:** Melanie Ernst, Tugba N. Ozturk, Janice L. Robertson.

**Funding acquisition:** Janice L. Robertson.

**Investigation:** Melanie Ernst, Tugba N. Ozturk, Janice L. Robertson.

**Methodology:** Melanie Ernst, Tugba N. Ozturk, Janice L. Robertson.

**Project administration:** Melanie Ernst, Janice L. Robertson.

**Resources:** Melanie Ernst, Tugba N. Ozturk, Janice L. Robertson.

**Software:** Melanie Ernst, Tugba N. Ozturk, Janice L. Robertson.

**Supervision:** Janice L. Robertson.

**Validation:** Melanie Ernst, Tugba N. Ozturk, Janice L. Robertson.

**Visualization:** Melanie Ernst, Tugba N. Ozturk, Janice L. Robertson.

**Writing – original draft:** Melanie Ernst, Tugba N. Ozturk, Janice L. Robertson.

**Writing – review & editing:** Melanie Ernst, Tugba N. Ozturk, Janice L. Robertson.

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
