## [Decision Letter · Decision Letter 0]

14 Sep 2022

PONE-D-22-21829A single-molecule method for measuring fluorophore labeling yields for the study of membrane protein oligomerization in membranesPLOS ONE

Dear Dr.Janice Lee Robertson,

Thank you for submitting your manuscript to PLOS ONE. After careful consideration, we feel that it has merit but does not fully meet PLOS ONE’s publication criteria as it currently stands. Therefore, we invite you to submit a revised version of the manuscript that addresses the points raised during the review process.

We look forward to receiving your revised manuscript.

Kind regards,

Sabato D'Auria

Academic Editor

PLOS ONE

Journal Requirements:

"The Robertson lab is supported by the National Institute of General Medical Science, National Institutes of Health (R01GM120260). "

"ME, TNO and JLR were supported with a grant (R01GM120260) from the National Institutes of General Medical Sciences, National Institutes of Health. The funders had no role in the study design, data collection and analysis, decision to publish, or preparation of the manuscript."

3. We noted in your submission details that a portion of your manuscript may have been presented or published elsewhere. 

"This study describes a novel analysis approach that is applied to data previously published in Chadda et al., JGP 2018 and Chadda et al., eLife 2016."

Reviewers' comments:

Reviewer's Responses to Questions

**Comments to the Author**

1. Is the manuscript technically sound, and do the data support the conclusions?

Reviewer #1: Yes

Reviewer #2: Yes

2. Has the statistical analysis been performed appropriately and rigorously? 

Reviewer #1: Yes

Reviewer #2: Yes

3. Have the authors made all data underlying the findings in their manuscript fully available?

Reviewer #1: Yes

Reviewer #2: Yes

4. Is the manuscript presented in an intelligible fashion and written in standard English?

Reviewer #1: Yes

Reviewer #2: Yes

5. Review Comments to the Author

Reviewer #1: This interesting study addresses a very important point, which is the quantification of fluorophore labeling, in experiments involving protein oligomerization within biological membranes. This methodology might considerably improves the accuracy of the description (i.e. the determination of the dissociation constant) of protein oligomerization in lipid bilayers.

Certainly, the application of this method is not very easy and presents some limits, but I think that the authors have carefully addressed this point in the introduction and in the discussion sections.

The high quality of the analysis presented for the case study (homodimeric CLC-ec1) makes this paper certainly suitable for publication in PLOS ONE.

Reviewer #2: The manuscript by Melanie Ernst et al. entitled “A single-molecule method for measuring fluorophore labeling yields for the study of membrane protein oligomerization in membranes” reports the development of a method to determine fluorophore labeling yields from single-molecule photobleaching data. The authors, based on them previous work with CLC-ec1 (single-molecule photobleaching subunit capture approach) developed a new method to evaluate the fluorophore labelling yields, the stoichiometry and KD values for a range of other CLC-ec1 membrane proteins. The authors combined a single-molecule photobleaching TIRF microscope measures with a computational method to predict the labeling yields. This work demonstrates an approach to measuring the fluorescent labeling yield directly from an experiment in real-time using a known dimeric control, representing a significant advance in the ability to determine fixed or dynamic stoichiometry for membrane proteins in membranes. The authors declare also some constrains: “the fluorophore labelling yields can be determined accurately with single-molecule amounts of well-established dimer controls greatly simplifying the quantitative requirements to study dynamic protein oligomerization”

The aim of this work is interesting (and this reviewer appreciates it), the paper is well done in almost all parts.

The developed method and experiments performed are interesting (and this reviewer appreciates it), the protocol is well described in all parts.

The manuscript in the present form demands a light revision before it can be published, so this reviewer suggests a minor revision of the paper.

The paper should be modified in some parts.

Major issues:

1) The abstract is too much long and results not clear, please reduce and improve.

2) Please adds the errors as bar in all graphs reported (i.e., Fig. 1 panel C is missed),

3) Please adds (in materials and methods) a short paragraph that describe the statistical analysis used,

4) Please divide the Discussion section in two sections (Discussion and Conclusion)

5) Which is the operational range of your methods? Please explicit the constrains and limits in the conclusion

Minor issue:

1) Please revise the text for some misspelling

6. PLOS authors have the option to publish the peer review history of their article (what does this mean?). If published, this will include your full peer review and any attached files.

Reviewer #1: No

Reviewer #2: No

---

## [Author Response · Author response to Decision Letter 0]

1 Dec 2022

We thank the reviewers for their considerate feedback and have revised the proposal accordingly. We believe the updated manuscript is significantly improved. Here is a detailed summary of our responses: 

Reviewer #1: This interesting study addresses a very important point, which is the quantification of fluorophore labeling, in experiments involving protein oligomerization within biological membranes. This methodology might considerably improves the accuracy of the description (i.e. the determination of the dissociation constant) of protein oligomerization in lipid bilayers.

Certainly, the application of this method is not very easy and presents some limits, but I think that the authors have carefully addressed this point in the introduction and in the discussion sections.

The high quality of the analysis presented for the case study (homodimeric CLC-ec1) makes this paper certainly suitable for publication in PLOS ONE.

Reviewer #2: The manuscript by Melanie Ernst et al. entitled “A single-molecule method for measuring fluorophore labeling yields for the study of membrane protein oligomerization in membranes” reports the development of a method to determine fluorophore labeling yields from single-molecule photobleaching data. The authors, based on them previous work with CLC-ec1 (single-molecule photobleaching subunit capture approach) developed a new method to evaluate the fluorophore labelling yields, the stoichiometry and KD values for a range of other CLC-ec1 membrane proteins. The authors combined a single-molecule photobleaching TIRF microscope measures with a computational method to predict the labeling yields. This work demonstrates an approach to measuring the fluorescent labeling yield directly from an experiment in real-time using a known dimeric control, representing a significant advance in the ability to determine fixed or dynamic stoichiometry for membrane proteins in membranes. The authors declare also some constrains: “the fluorophore labelling yields can be determined accurately with single-molecule amounts of well-established dimer controls greatly simplifying the quantitative requirements to study dynamic protein oligomerization”

The aim of this work is interesting (and this reviewer appreciates it), the paper is well done in almost all parts.

The developed method and experiments performed are interesting (and this reviewer appreciates it), the protocol is well described in all parts.

The manuscript in the present form demands a light revision before it can be published, so this reviewer suggests a minor revision of the paper.

The paper should be modified in some parts.

Major issues:

1) The abstract is too much long and results not clear, please reduce and improve.

Thank you for pointing this out. We have revised the abstract to elaborate on the results to hopefully make this clearer. In addition, the abstract is now shortened to 237 words, adhering to the journal guidelines. 

2) Please adds the errors as bar in all graphs reported (i.e., Fig. 1 panel C is missed),

We revised the figures to improve visibility of error bars that were not visible in the previous version. Originally, we did not include error bars in Fig 1C because these plots reflect model data derived from a mathematical simulation. However, since this is a stochastic model where we randomly simulate protein partitioning into liposomes, we can report the variability associated with the results, even though the standard deviation in the data are small. We have now modified Fig 1C to include these error bars that reflect the simulation variability.

3) Please add (in materials and methods) a short paragraph that describe the statistical analysis used,

In the “Materials and methods” section, we have added a final section titled "Statistical analyses" that describes the type of errors calculated throughout, and the statistical tests that we used (page 12-13, lines 322-459). In addition, we modified Fig 4C to report mean ± sem from independent KD estimations per sample, rather than the standard deviation of the parameter estimation. The new representation allows for statistical testing between means that will be more useful for readers of this study. 

4) Please divide the Discussion section in two sections (Discussion and Conclusion)

The discussion is now divided into "Discussion" and "Conclusion" sections. 

5) Which is the operational range of your methods? Please explicit the constrains and limits in the conclusion

This is an excellent question and we thank the reviewer for asking this as we neglected to include this important result in the previous version of our manuscript. We have now revised the paper to include a new Fig 5, which shows a statistical analysis of the operational range for these studies based on the ability to discriminate monomer and dimer photobleaching probability distributions. With this, we conclude that for mole fraction densities of *χ* = 10-9 to 10-5 subunits/lipid, that Pfluor � 0.7 and Pbg � 0.1 allow for significant discrimination between populations. However, if the reaction can be well described for *χ* � 10-6 subunits/lipid, then the labeling conditions can be lowered to Pfluor � 0.4, Pbg � 0.1. Despite this, we comment that labeling should be optimized as close to Pfluor to 1 and Pbg to 0 to increase robustness in this photobleaching approach. This analysis is now described in a new section of the results on page 21-22, lines 741-826, Fig 5, and in the “Conclusions” section on page 26, lines 1030-1036.

Minor issue:

1) Please revise the text for some misspelling

We have now carefully read through the manuscript and revised the text for misspelling.

---

## [Editor Report · Decision Letter 1]

6 Jan 2023

A single-molecule method for measuring fluorophore labeling yields for the study of membrane protein oligomerization in membranes

PONE-D-22-21829R1

Dear Dr. Janice Lee Robertson,

We’re pleased to inform you that your manuscript has been judged scientifically suitable for publication and will be formally accepted for publication once it meets all outstanding technical requirements.

Kind regards,

Sabato D'Auria

Academic Editor

PLOS ONE
---

## [Editor Report · Acceptance letter]

12 Jan 2023

PONE-D-22-21829R1 

A single-molecule method for measuring fluorophore labeling yields for the study of membrane protein oligomerization in membranes 

Dear Dr. Robertson:

I'm pleased to inform you that your manuscript has been deemed suitable for publication in PLOS ONE. Congratulations! Your manuscript is now with our production department. 

Kind regards, 

on behalf of

Dr. Sabato D'Auria 

Academic Editor

PLOS ONE